# Safety of bedside surgical tracheostomy during COVID-19 pandemic: A retrospective observational study

Edoardo Picetti[1]*, Anna Fornaciari[1], Fabio Silvio Taccone[2], Laura Malchiodi[1], Silvia Grossi[1], Filippo Di Lella[3], Maurizio Falcioni[3], Giulia D'Angelo[3], Emanuele Sani[1], Sandra Rossi[1]

**1** Department of Anesthesia and Intensive Care, Parma University Hospital, Parma, Italy, **2** Department of Intensive Care, Erasme Hospital, Université Libre de Bruxelles, Brussels, Belgium, **3** Department of Otolaryngology, Parma University Hospital, Parma, Italy

\* edoardopicetti@hotmail.com

**Data Availability Statement:** All relevant data are within the manuscript.

**Funding:** The authors received no specific funding for this work.

## Abstract

Data regarding safety of bedside surgical tracheostomy in novel coronavirus 2019 (COVID-19) mechanically ventilated patients admitted to the intensive care unit (ICU) are lacking. We performed this study to assess the safety of bedside surgical tracheostomy in COVID-19 patients admitted to ICU. This retrospective, single-center, cohort observational study (conducted between February, 23 and April, 30, 2020) was performed in our 45-bed dedicated COVID-19 ICU. Inclusion criteria were: a) age over 18 years; b) confirmed diagnosis of COVID-19 infection (with nasopharyngeal/oropharyngeal swab); c) invasive mechanical ventilation and d) clinical indication for tracheostomy. The objectives of this study were to describe: 1) perioperative complications, 2) perioperative alterations in respiratory gas exchange and 3) occurrence of COVID-19 infection among health-care providers involved into the procedure. A total of 125 COVID-19 patients were admitted to the ICU during the study period. Of those, 66 (53%) underwent tracheostomy. Tracheostomy was performed after a mean of 6.1 (± 2.1) days since ICU admission. Most of tracheostomies (47/66, 71%) were performed by intensivists and the mean time of the procedure was 22 (± 4.4) minutes. No intraprocedural complications was reported. Stoma infection and bleeding were reported in 2 patients and 7 patients, respectively, in the post-procedure period, without significant clinical consequences. The mean $PaO_2 / FiO_2$ was significantly lower at the end of tracheostomy (117.6 ± 35.4) then at the beginning (133.4 ± 39.2) or 24 hours before (135.8 ± 51.3) the procedure. However, $PaO_2/FiO_2$ progressively increased at 24 hours after tracheostomy (142 ± 50.7). None of the members involved in the tracheotomy procedures developed COVID-19 infection. Bedside surgical tracheostomy appears to be feasible and safe, both for patients and for health care workers, during COVID-19 pandemic in an experienced center.

**Competing interests:** NO authors have competing interests

## Introduction

The novel coronavirus 2019 (COVID-19) is associated with a respiratory disease ranging from mild cough to severe acute respiratory distress syndrome (ARDS) [1, 2]. Many patients worldwide, requiring invasive mechanical ventilation, has been admitted to the intensive care units (ICUs) until now [3–7]. Tracheostomy (surgical or percutaneous) is generally considered when the need for mechanical ventilation is expected to be prolonged [8, 9]. Many potential benefits associated with this procedure has been described in general ICU patients, such as improved patient comfort, less need for sedation and faster weaning from mechanical ventilation [8, 9]. However, tracheostomy is an aerosol-generating procedure and several concerns in COVID-19 patients on to its safety has been raised [10–12]. Despite some studies published during COVID-19 pandemic [13–20], more data are necessary to better elucidate the role of tracheostomy in this setting. The aim of this study was therefore to evaluate the safety of bedside surgical tracheostomy in COVID-19 patients admitted to ICU.

## Materials and methods

After Institutional Review Board (IRB) approval (Comitato Etico AVEN 562/2020/OSS/AOUPR), we conducted a single-center retrospective analysis of bedside surgical tracheotomies performed in our 45-bed dedicated COVID-19 ICU between February, 23 and April, 15, 2020. Written informed consent was obtained. Inclusion criteria were: a) age over 18 years; b) confirmed diagnosis of COVID-19 infection (with nasopharyngeal/oropharyngeal swab); c) invasive mechanical ventilation and d) clinical indication for tracheostomy. Surgical tracheostomy was generally performed in all salvageable COVID-19 patients without coagulation disorders (coagulation parameters and platelets count within the normal range) by 2 ICU physicians or 2 otolaryngologists; also, one anesthesiologist (generally a member of the ICU staff) and one ICU nurse were present during the procedure. Surgical tracheostomies were performed according to the Parma tracheostomy technique [21, 22]. For general anesthesia induction we have utilized propofol (2 ml/kg), fentanyl (1.5 mcg/kg) and rocuronium (0.6 mg/kg); for anesthesia maintenance, a continuous infusion of propofol (4–5 mg/kg/h) was started after induction. Adjunctive fentanyl boluses are administered as required in case of arterial hypertension and/or tachycardia. The patient was positioned flat, with gentle hyperextension of the head and neck. The fraction of inspired oxygen (FiO$_2$) was increased to reach the value of 100%. Thereafter, a 2 cm median, vertical incision was performed about 1 cm below the cricoid cartilage. Soft tissues were then bluntly dissected with a mosquito-clamp and with the help of two Volkmann's retractors. The thyroid isthmus was generally retracted upwards. Hemostasis was obtained either mechanically (by compression with sterile gauzes) or by using a bipolar coagulation device. The blunt dissection was continued in a deep-ward direction until reaching the trachea at the level of the first-second (or second-third) tracheal ring. The tracheal tube was then pushed slightly forward (without deflating the cuff and turning off the ventilator), in order to place its cuff below the site of incision. The mechanical ventilation was shortly paused, the tracheal tube cuff deflated (to minimize the risk of rupture) and the trachea was incised between the first and the second or the second and the third ring; the cuff was then immediately re-inflated with resumption of the mechanical ventilation. The incision was dilated with a Laborde's forceps or a Killian speculum. During the incision of the trachea, the aspirator was kept close to the cut in order to drain secretions and droplet reducing contamination. The appropriate tracheostomy cannula was lubricated and its lumen was loaded with a blunt-tip soft silicone catheter acting as a guidewire during the insertion process, preventing possible posterior tracheal wall damage. Ventilation was then shortly paused (mechanical ventilation in stand-by without air flow for maximum near 5 seconds) and the tracheal tube was

retracted above the tracheal incision. The tracheostomy cannula was inserted into the trachea and, after ensuring its correct positioning, ventilation was restarted through it. Recruitment manoeuvres were utilized at the end of the procedure if clinically indicated. Every member of the team had personal protective equipment (PPE) composed of: surgical cap, N95 mask (FFP2), face shield, a double pair of non-sterile gloves, full body gown and waterproof boot. The surgical team was also equipped with sterile surgical gloves and gown. The rooms of the COVID-19 ICU were not equipped with negative pressure systems. Powered Air Purifying Respirators (PAPRs) were not available.

We retrieved from clinical records the following data (last access July 15, 2020): age, sex, simplified acute physiology score (SAPS) II at ICU admission, the rate of lung involvement (thorax CT scan) at ICU admission, arterial partial pressure of oxygen ($PaO_2$) / inspired fraction of oxygen ($FiO_2$) on ICU admission, ICU length of stay (LOS), duration of invasive mechanical ventilation and outcome at ICU discharge (i.e. death or alive). For each tracheostomy we retrieved: the time from ICU admission to tracheostomy (days), the tracheostomy procedure duration (minutes), the presence of tracheostomy at ICU discharge and procedure related complications (intra-operative and post-operative), such as tracheal ring fracture, stoma bleeding (requiring surgical interventions and/or topical hemostatic agents) and/or infections, pneumothorax, cannula dislodgement and death caused by the procedure. Moreover, we collected positive end-expiratory pressure (PEEP), $FiO_2$, arterial partial pressure of oxygen $PaO_2$), arterial partial pressure of carbon dioxide ($PaCO_2$), pH, arterial oxygen saturation ($SatO_2$) and arterial blood lactate registered at different time points: 24 hours before tracheostomy (T1), immediately before the procedure (T2), immediately after tracheostomy (T3) and 24 hours after tracheostomy (T4). Follow-up was until April 30, 2020. Moreover, until this date, clinical conditions of members involved in the tracheostomy procedures were also monitored.

The objectives of this study were to describe, during the ICU stay: 1) perioperative complications, 2) perioperative alterations in respiratory gas exchange and 3) occurrence of COVID-19 infection among health-care providers involved into the procedure.

### Statistical analysis

Continuous data are expressed as mean and standard deviation (SD); ordinal data, such as scoring evaluation, as median and interquartile range (IQR); categorical data as number and percentage. Skewness and Kurtosis test was applied to test normality of the distribution of the continuous variables and Levene's test for homogeneity of variances. Comparison were made for ventilator (PEEP and $FiO_2$), blood gas-analysis (pH, $PaO_2$, $PaCO_2$, $PaO2/FiO_2$, $SatO_2$) and metabolic (lactates) parameters at four different time points respect to the tracheostomy procedure: 24 h before, at the start, at the end, and 24 h after. Due to the non-normal distribution of the variables as well as of the residuals from the analysis of variance, the non-parametric Friedman rank sum test for repeated measures was employed for the statistical analysis. Post-hoc analysis was done with the Wilcoxon matched-pairs signed-ranks test with Bonferroni correction for multiple comparisons. Two tailed P-value less than 0.05 was considered to be statistically significant. Data were analysed using STATA v. 13.0 (College Station, TX).

### Results

A total of 125 COVID-19 patients were admitted to the ICU during the study period. Of those, 66 (53%) underwent tracheostomy. Patients characteristics are summarized in Table 1.

The mean age was 58.7 (± 8.7) years and mostly were males (n = 54; 82%). Median SAPS II on ICU admission was 45 (IQR 43–50). The median percentage of lung involvement at the

**Table 1. Baseline characteristics of the patient population.**

| | |
|---|---|
| No. of subjects | 66 |
| Age, mean (SD), y | 58.7 (8.7) |
| Sex | |
| Male, no (%) | 54 (82) |
| Female, no (%) | 12 (18) |
| SAPS II*, median (IQR) | 45 (43–50) |
| Thorax CT Score*, median (IQR) | 50 (40–65) |
| $PaO_2/FiO_2$*, mean (SD) | 97.9 (47.1) |
| Duration of IMV, mean (SD), days | 21.4 (7.2) |
| ICU LOS, mean (SD), days | 23.2 (6.9) |
| Outcome ICU** | |
| Alive, no (%) | 57 (86.4) |
| Dead, no (%) | 9 (13.6) |
| 18 in ICU | |

Abbreviations: No. = number, SD = standard deviation, y = years, SAPS = Simplified Acute Physiology Score, IQR = interquartile range, CT = computed tomography, $PaO_2$ = arterial partial pressure of oxygen, $FiO_2$ = fraction of inspired oxygen, IMV = invasive mechanical ventilation, ICU = intensive care unit, LOS = length of stay.

* = at ICU admission.

** = at ICU discharge.

thorax computed tomography (CT) was 50 (40–65) % while the mean $PaO_2/FiO_2$ on ICU admission was 97.9 (± 47.1). The mean duration of invasive mechanical ventilation was 21.4 (± 7.2) days; the time from ICU admission to the tracheostomy was 6.1 (± 2.1) days. At the last day of follow-up, 39 (59.1%) patients were discharged alive from ICU, while 18 (27.3%) were still in the ICU.

Tracheostomy data are reported in Table 2.

Almost 71.2% of tracheostomies were performed by intensivists and the mean time of the procedure was 22 (± 4.4) minutes. Mean $PaO_2/FiO_2$ at the moment of tracheostomy was 133.4

**Table 2. Tracheostomy data.**

| | |
|---|---|
| Timing of tracheostomy*, mean (SD), days | 6.1 (2,1) |
| Duration of tracheostomy, mean (SD), min | 22 (4.4) |
| Surgeon's specialty, no (%) | |
| ICU physician | 47 (71.2) |
| ORL physician | 19 (28.8) |
| Intra-procedural complications, no (%) | |
| None | 66 (100) |
| Post-operative complications, no (%) | |
| Stoma infection | 2 (3.0) |
| Bleeding | 7 (10.6) |
| None | 57 (86.4) |
| Tracheostomy at the ICU discharge, no (%) | |
| Present | 39 (59.1) |
| No yet discharged | 18 (27.3) |

Abbreviations: no = number, ICU = intensive care unit, ORL = otolaryngology, SD = standard deviation.

* = from ICU admission.

**Table 3.  Respiratory parameters before and after tracheostomy.**

|  | 24 h before | start | end | 24 h after | P |
|---|---|---|---|---|---|
| PEEP, cmH$_2$O | 11.5±1.6 | 11.4±1.6 | 11.4±1.9 | 11.6±2.1 | 0.844 |
| FiO$_2$, % | 65.9±14.3 | 64.6±12.6 | 70.0±11.9 | 67.3±14.7 | 0.010 |
| pH | 7.4±0.1 | 7.4±0.1 | 7.4±0.1 | 7.4±0.1 | 0.582 |
| PaCO$_2$, mmHg | 51.3±10.7 | 54.8±14.8 | 53.5±11.2 | 51.8±9.0 | 0.508 |
| PaO$_2$, mmHg | 86.2±23.9 | 84.4±17.4 | 79.8±18.1 | 91.5±26.8 | 0.106 |
| PaO$_2$/FiO$_2$ | 135.8±51.3 | 133.4±39.2 | 117.6±35.4*† | 142.0±50.7‡ | 0.013 |
| SatO$_2$, % | 94.9±2.6 | 94.8±2.9 | 94.0±3.3 | 95.1±3.3 | 0.165 |
| Lactate, mmol/L | 1.5±0.6 | 1.4±0.5 | 1.5±0.6 | 1.5±0.5 | 0.424 |

Abbreviations: PEEP = positive end-expiratory pressure, FiO$_2$ = fraction of inspired oxygen, PaCO$_2$ = arterial partial pressure of carbon dioxide, PaO$_2$ = arterial partial pressure of oxygen, SatO$_2$ = arterial oxygen saturation.

˚ = data are reported as a mean ± standard deviation (SD).

* = P<0.05 vs. 24 h before

† = P<0.05 vs. start

‡ = P<0.05 vs. end.

(±39.2). None intra-procedural complications was reported. Considering post-procedural complications, we reported stoma infection in 2 patients, which was treated with topical antibiotics, and bleeding in 7 patients (i.e. needing surgical revision in 3 cases and topical hemostatis in 4 cases; in all cases anticoagulation was temporarily stopped). No blood transfusion was required after the procedure. No patient died because or immediately after the procedure. At ICU discharge, all patients (39/39) still had tracheostomy in place as well as all remaining ICU patients (18/18). Considering the latter, at the moment of the article submission, 12 patients were discharged alive from ICU (with tracheostomy in place) while 6 patients died in the ICU. All patients discharged alive from ICU has been decannulated in the intermediate care unit or in the rehabilitation department.

Table 3 reports the respiratory parameters at different time points with regard to the tracheostomy.

Only the PaO$_2$/FiO$_2$ significantly changed after tracheostomy. In particular, the mean PaO$_2$/FiO$_2$ at T3 (117.6 ± 35.4) was lower than T2 (133.4 ± 39.2) and T1 (135.8 ± 51.3). However, the mean PaO$_2$/FiO$_2$ recorded at T4 (142 ± 50.7) was higher respect to T3 (Fig 1).

At the last day of follow-up, none of the members involved in the tracheotomy procedures developed COVID-19 symptoms. Nasopharyngeal swab and serology were negative in all of them.

## Discussion

This study suggests that surgical bedside tracheostomy, performed with an adequate PPE, is a safe procedure both for the COVID-19 patients and for health-care workers. To our knowledge, this represents the largest study on this topic (bedside surgical tracheostomy) in COVID-19 patients in a single institution.

In our population, tracheostomy was performed on average 6.1 days after ICU admission. Our procedures can be considered as an "early" tracheostomy. The optimal timing of tracheostomy, which has been investigated in several studies on critically ill patients, remains a very controversial topic [23, 24]. Possible beneficial effects for early procedure can be considered the reduction in the rate of ventilator-associated pneumonia (VAP) and the lower requirement for sedatives [23, 24]. In our daily ICU clinical practice, early tracheostomy was the standard

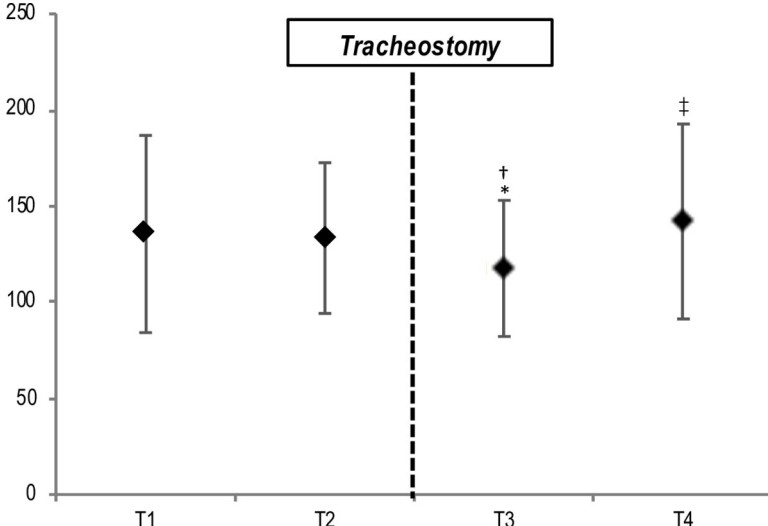

**Fig 1. PaO$_2$/FiO$_2$ at different time point respect to tracheostomy.** Abbreviations: T1 = 24 hours before tracheostomy, T2 = immediately before tracheostomy, T3 = immediately after tracheostomy, T4 = 24 hours after tracheostomy. * = P<0.05 vs. 24 h before; † = P<0.05 vs. start; ‡ = P<0.05 vs. end.

of care even before the COVID-19 pandemic. Therefore, we decided to continue our practice because it was expected to have a long time of mechanical ventilation and, by improving patients' comfort, reducing the need for sedation and providing a faster weaning from mechanical ventilation [8, 9]. Moreover, in our hospital, we created "intermediate care" units, which were skilled in the weaning management of tracheostomized patients; this represents an important issue for a rapid access to the post-acute care and also to have more ICU beds available for upcoming patients. It's possible that with this aggressive strategy some patients may have lost the opportunity of weaning without tracheostomy. However, all enrolled patients, with regard to the PaO$_2$/FiO$_2$ and the extent of lung involvement, were severely ill. As such, most of those would have spent a prolonged time on invasive mechanical ventilation. Whether the an early timing would increase the risk of transmission because of the high viral load [14], it has not been confirmed as none of the participants had COVID-19.

We decided to perform surgical tracheostomies, instead to percutaneous tracheostomies, mainly for two reasons: 1) usually (i.e. pre COVID-19), surgical tracheostomy was typically used more frequently than the percutaneous one because we treat many patients with acute brain injury [25] and 2) to reduce aerosolization (i.e. standard percutaneous tracheostomy generally requires a more extensive airway manipulation) [10]. Considering this last point, a novel percutaneous technique has been recently proposed [16]. This approach, consisting in the placement of bronchoscope alongside the endotracheal tube (and not inside it), was utilized in 270 patients and appeared safe and effective for patients and health care workers. In this regard, none of the health-care workers involved in the tracheostomy procedures developed COVID-19 infection, even in the absence of a negative pressure environment and PAPRs. This aspect was investigated only in the personnel involved in the procedure. In our ICU, during the COVID period, we have only 4 doctors and 3 nurses with confirmed COVID-19 disease (mild forms). None of these was involved in the tracheostomy procedures. Moreover, we performed bedside procedures without the need to move patients in the operating room (OR). This aspect is cost-effective and also associated with less risk for the patients and health-care workers during the pandemic [10]. We have observed an absence of intra-operative complications and a low incidence of post-operative complications (mainly minor

bleeding). This last could be linked to the resumption of low molecular weight heparin (50–100 UI/kg of enoxaparin twice daily), which was given in all patients, considering the elevated risk of venous thromboembolism [26]. Moreover, our technique seems to be safe also because of the absence of clinically significant alterations on gas exchanges; this finding could be related to the interruption of ventilation only for a few seconds during the procedure.

Several articles regarding tracheostomy in COVID-19 patients has been published [13–20]. Martin-Villares et al. [13] performed a multicenter prospective observational study of 1890 COVID-19 patients undergoing tracheostomy (81.3% surgical and 22.7% percutaneous) performed by otolaryngologists across 120 hospitals in Spain. Tracheostomy was performed after a median of 12 (4–42) days since orotracheal intubation. Most procedures were performed at bedside in the ICU and PARPS were available only in 2 hospitals. A low rate of complications has been reported. Chao et al. [15] performed a prospective single-system multicentre observational cohort study (5 hospitals within the University of Pennsylvania Health System) of 53 COVID-19 patients who underwent tracheostomy (29 percutaneous and 24 surgical) performed by 11 different attending surgeons. The mean time from endotracheal intubation to tracheostomy was 19.7 ($\pm$ 6.9) days. PARPs were used in 30 cases. All procedures, except 1 done in the OR, were performed at bedside in the ICU in a negative pressure room. No COVID-19 cases related to the procedures among health care workers were reported. Only few minor complications were registered. Piccin et al. [17] described 24 surgical bedside tracheostomies on COVID-19 patients performed by otolaryngologists. The median timing of tracheostomy was 10 days after intubation. PARPs were always utilized. No complications were reported. Mattioli et al. [19] reported 28 tracheostomies (surgical and percutaneous) on COVID-19 ICU patients without significant complications; some procedures were performed within 7 days from intubation without reported infections among proceduralists. Volo et al. [20] performed a retrospective cohort study on 23 COVID-19 patients undergoing ICU bedside tracheostomy (22 surgical and 1 percutaneous) performed by otolaryngologists. The average time between the intubation date and the tracheostomy date was 13 days. All the procedures were performed in negative pressure-rooms. None of the tracheostomy team developed COVID-19 disease. No significant complications were reported.

Our study, respect to those described above, has some peculiarities: 1) the majority of the bedside surgical tracheostomies were performed by trained intensivists with a low rate of complications and 2) the timing of tracheostomy was earlier without negative effects on gas exchange (perioperative data about this aspect are not reported in published studies) and safety (no infections among proceduralists) despite the lack of PAPRs and negative pressure rooms.

Our study presents several limitations to acknowledge. First, the retrospective design might have underestimated episodes of desaturation during and after the procedure. Also, the uncomplete follow-up might provide a misleading quantification of poor outcome among these patients. As our center has a long-lasting experience with this procedure, findings could not be easily extrapolated to other centers. Data about the timing of decannulation are not available. This study was focused on the ICU period. Moreover, this study was not designed to assess whether tracheostomy could be associated to better outcomes in this setting; as such, larger and well-designed prospective studies are necessary.

In conclusions, bedside surgical tracheostomy appears to be feasible and safe, both for patients and for health care workers, during COVID-19 pandemic in an experienced center.

## Author Contributions

**Conceptualization:** Edoardo Picetti, Fabio Silvio Taccone, Emanuele Sani.

**Data curation:** Edoardo Picetti, Anna Fornaciari, Silvia Grossi, Sandra Rossi.

**Formal analysis:** Edoardo Picetti.

**Methodology:** Edoardo Picetti, Filippo Di Lella.

**Software:** Laura Malchiodi, Giulia D'Angelo.

**Supervision:** Fabio Silvio Taccone, Maurizio Falcioni.

**Writing – original draft:** Edoardo Picetti.

**Writing – review & editing:** Anna Fornaciari, Fabio Silvio Taccone, Laura Malchiodi, Silvia Grossi, Filippo Di Lella, Maurizio Falcioni, Giulia D'Angelo, Emanuele Sani, Sandra Rossi.

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
