## [Decision Letter · Decision Letter 0]

19 Aug 2020

PONE-D-20-24276

Safety of bedside surgical tracheostomy during COVID-19 pandemic a retrospective observational study

PLOS ONE

Dear Dr. Picetti,

Thank you for submitting your manuscript to PLOS ONE. After careful consideration, we feel that it has merit but does not fully meet PLOS ONE’s publication criteria as it currently stands. Therefore, we invite you to submit a revised version of the manuscript that addresses the points raised during the review process.

We look forward to receiving your revised manuscript.

Kind regards,

Corstiaan den Uil

Academic Editor

PLOS ONE

Journal Requirements:

2. Thank you for including your ethics statement:  "Comitato Etico AVEN 562/2020/OSS/AOUPR written informed consent"

Please amend your current ethics statement to confirm that your named institutional review board or ethics committee specifically approved this study.

3. Please include the date(s) on which you accessed the databases or records to obtain the data used in your study.

"NO"

Reviewers' comments:

Reviewer's Responses to Questions

**Comments to the Author**

1. Is the manuscript technically sound, and do the data support the conclusions?

Reviewer #1: Yes

Reviewer #2: Yes

2. Has the statistical analysis been performed appropriately and rigorously? 

Reviewer #1: N/A

Reviewer #2: Yes

3. Have the authors made all data underlying the findings in their manuscript fully available?

Reviewer #1: Yes

Reviewer #2: Yes

4. Is the manuscript presented in an intelligible fashion and written in standard English?

Reviewer #1: Yes

Reviewer #2: Yes

5. Review Comments to the Author

Reviewer #1: This is a very interesting work of a long series from a single centre in one of the hot-spots of the pandemic in Europe: the North of Italy. In particular it focusses in evaluating the safety of bedside 67 surgical tracheostomy in COVID-19 patients admitted to ICU. The results of no contagion after 65 tracheostomies without negative pressure room, a very good result: congratulations.

In my opinion the manuscript can be published after considering some comments

Reviewer #2: Manuscript Number: PONE-D-20-24276

Full Title: Safety of bedside surgical tracheostomy during COVID-19 pandemic a retrospective observational study

Summary: This is a retrospective observational study of bedside surgical tracheostomy performed on COVID-19 patients in Italy during the pandemic. 66 patients underwent tracheostomy, out of 125 admitted during this period. Mean time to tracheostomy was 6 days. Mean FiO2%/ PEEP prior to tracheostomy were 65%/11. All procedures were completed successfully. Complications were few, with 3% stomal infection and 10% bleeding. Mortality was 14% overall. N95s were used for PPE and no proceduralists were infected.

Pros: Useful case series that demonstrates that early surgical bedside trach is feasible in COVID-19 with a low rate of complications, that ICU mortality is low in these patients and that N95 masks appear to afford adequate protection for what is sometimes considered a high transmission-risk procedure.

Cons: Retrospective, single center. Follow-up truncated at ICU discharge. Stated objective is determination of safety, so some aspects need to be more clearly defined a-priori such as significant bleeding. Overall strategy toward tracheostomy in the patients would be useful to know.

Introduction: There are now many published studies describing tracheostomy in COVID-19, including a 1890 patient multicenter series from Spain (PMID 32749607). Please reference these and mention the additional value of this study.

Methods:

1. The primary objective appears to be determination of safety. Therefore, better a priori definition of safety endpoints and other complications is required.

a. Please define what was considered significant bleeding more clearly. I gather this was the need for surgical intervention or use of hemostatic agents, but please state explicitly in the Methods section. Important question- was oozing requiring temporary cessation of anticoagulation included in this definition? If not, why?

b. Was sustained oxygen desaturation considered a complication? Ideally, the authors should define sustained oxygen desaturation during and immediately after the procedure (SpO2< XX% for XX seconds/ minutes) and report this. If you did not have sufficient documentation to report this, please state as a limitation in the limitations paragraph.

c. Did the authors look at early dislodgment (within 5-7 days)? This is another complication related to choice of tube and sometimes position of tube.

d. Please list in the methods section all of the specific complications that you looked for.

2. A critical element of safety is transmission to healthcare providers. This is a very useful element of this study, particularly since the providers had no negative pressure rooms and used only N95s.

a. Please clarify that Powered Air Purifying Respirators (PAPR) were not used.

b. The authors state that no healthcare staff were infected. However, in my experience this is often loosely defined in studies. For example, do you mean just the proceduralists (intensivists, ENT surgeons, anesthesiologists and assistants) or do you include all nurses, technicians and respiratory therapists in the room. If you included all nurses etc, did you survey 100% of the nurses present in the ICU room during tracheostomy to confirm that none developed symptoms? This seems unlikely given this was retrospective study. Please clarify. If you only have information on proceduralists, please state this clearly, and separately mention as a limitation in the limitations paragraph.

3. More information on the overall protocol or strategy toward tracheostomy would be useful. Did the unit have a written protocol on timing, procedure, PPE etc?

4. Why was follow-up truncated at ICU discharge? Since this is a safety-focused study, I would expect information on complications (such as bleeding, infection, granuloma, early dislodgment, need for revision) at least for the duration of hospital admission.

5. Information on decannulation would be ideal, however, it is understandable if this is unavailable- please mention under limitations.

Results:

Please see the points I raise under Methods

Discussion:

1. There are now many published studies on tracheostomy performed for COVID-19. Just to mention a few of these- PMIDs 32749607, 32741194, 32709307, 32656673, 32541213 and 32339508. Please describe what this study provides that prior studies do not, and contrast your findings to theirs.

2. Please discuss why you did not wait 10-14 days, when viral load is lower and risk of transmission to healthcare providers lower?

6. PLOS authors have the option to publish the peer review history of their article (what does this mean?). If published, this will include your full peer review and any attached files.

Reviewer #1: **Yes: **Cristina Martin-Villares MD PhD

Reviewer #2: **Yes: **Venkatakrishna Rajajee

---

## [Author Response · Author response to Decision Letter 0]

16 Sep 2020

REVIEWER 1

1. In line 191, the authors reported that the median time from ICU admission to tracheostomy was 6.1 (± 2.1) days. With such an early strategy, more than 50% of critical intubation patients underwent tracheostomy. This is a very controversial topic, the timing of tracheostomy in intubated patients, and trials are impossible during COVID-19 pandemic. But with this aggressive strategy some patients could loose the opportunity of weaning without tracheostomy? 

Authors’ response: Many thanks for this important comment. Data about optimal timing of tracheostomy are lacking in COVID-19 patients. It’s possible that with this aggressive strategy some patients may have lost the opportunity of weaning without tracheostomy. However, all patients enrolled, considering the PaO2/FiO2 and the percentage of lung involvement, were very ill. For this reason, it was difficult to think to a short time of invasive mechanical ventilation. Moreover, there are other 2 aspects to consider: 1) early tracheostomy was the standard in our ICU also before the COVID-19 pandemic and 2) we created “intermediate care” units, skilled in the weaning management of tracheostomized patients; this represents an important issue for a rapid access to the post-acute care and also to have more ICU beds available for upcoming patients. We have added a comment regarding this aspect in the discussion section.

2. In line 191-192, investigators reported that at the last day of follow-up, 39 (59.1%) patients were discharged alive from ICU, while 18 (27.3%) were still in the ICU. This successfully results needs more follow up with 27,3% of patients still under mechanical ventilation with tracheostomy. 

Authors’ response: At the moment of the article submission to PLOSONE, 12 patients has been discharged alive from ICU (with tracheostomy in place) while 6 patients died in the ICU. We have added this data in the results section. 

3. In line 217, I can read that 71.2 % of tracheostomies were performed by intensivists. In Spain, open tracheostomies are performed by otolaryngologists (or sometimes thoracic and maxillofacial surgeons) and intensivists only perform percutaneous tracheostomies. 

Authors’ response: We are aware that this is a characteristic of our center. In particular, intensivists have been doing surgical bedside tracheostomy for about 40 years. Some of us do about 30-40 surgical tracheostomies every year. For this reason, among the limitations of the study, we stated: "As our center has a long-lasting experience with this procedure, findings could not be easily extrapolated to other centres”.

4. In lines 268-269, “To our knowledge, this 269 represents the largest study on this topic (bedside surgical tracheostomy) in COVID-19 patients” in A SINGLE INSTITUTION. A recent paper published in European Archives of Oto-Rhino-Laryngology (Outcome of 1890 tracheostomies for critical COVID‑19 patients: a national cohort study in Spain (https://doi.org/10.1007/s00405-020-06220-3) describes a larger number of tracheostomies in various institutions of our group) 

Authors’ response: We agree with the reviewer. Congratulations for this recent published paper. As suggested, we added the term “in a single institution” to the sentence.

5. We are agree in the low rate of complications in COVID-19 tracheostomies in our experience.

Authors’ response: We thank the reviewer for this comment. 

6. In line 223-24, the investigators report that at ICU discharge, all patients (39/39) still had tracheostomy in place. I understand that there were no patients decannulated from 65 tracheostomies. In our experience, weaning is complex, but decannulation after weaning is almost the rule. 

Authors’ response: We thank the reviewer for this comment. At the moment of the article submission to PLOSONE, all patients discharged alive from the ICU has been decannulated outside of the ICU (generally in the intermediate care unit or in the rehabilitation department). Unfortunately, we did not collect this data (ex. timing of decannulation). We have added this information in the results section and as limitation.

7. In line 283-285, the reasons why the authors decided to perform surgical tracheostomies, instead of percutaneous tracheostomies are controversial. 

Authors’ response: As discussed in the article, we perform surgical tracheostomies, instead to percutaneous, for 2 reasons:

1 - in our department, also in the pre-COVID period, we done more surgical tracheostomies (near 80%) respect to percutaneous (near 20%). We are more familiar with this type of technique because we treat, every year, many patients with acute brain injuries; in this category of patients surgical tracheostomy is better tolerated.

2 – to reduce aerosolization. Standard percutaneous tracheostomy, excluding the particular technique utilized by Angel L et al. (Ann Thor Surgery 2020; 110: 1006-1011), generally requires a more extensive airway manipulation.

We have added this latest article to the discussion section.

8. In Bibliography, I recommended to review another Italian series in discussion as: 

Picin University of Bologna 24 tracheostomies

DÁsciano Santa Croce Pesaro 22 tracheostomies

Mattioli University of Modena 28 tracheostomies

Authors’ response: As requested, we have added this article to the discussion section.

REVIEWER 2

1. Introduction: There are now many published studies describing tracheostomy in COVID-19, including a 1890 patient multicenter series from Spain (PMID 32749607). Please reference these and mention the additional value of this study.

Authors’ response: As requested, we have referenced recently published studies mentioning the additional value of this study

2. The primary objective appears to be determination of safety. Therefore, better a priori definition of safety endpoints and other complications is required.

a. Please define what was considered significant bleeding more clearly. I gather this was the need for surgical intervention or use of hemostatic agents, but please state explicitly in the Methods section. Important question- was oozing requiring temporary cessation of anticoagulation included in this definition? If not, why?

Authors’ response: Many thanks for this comment. Significant bleeding was defined was a bleeding requiring surgical intervention or use of topical hemostatic agents. We have added this statement in the methods section. In all 7 cases of postoperative bleeding, anticoagulation was temporarily stopped. In our daily clinical practice, if oozing requires a temporary cessation of anticoagulation, we also apply topical hemostatic agents always.

3. b. Was sustained oxygen desaturation considered a complication? Ideally, the authors should define sustained oxygen desaturation during and immediately after the procedure (SpO2< XX% for XX seconds/ minutes) and report this. If you did not have sufficient documentation to report this, please state as a limitation in the limitations paragraph.

Authors’ response: Many thanks for this comment. We don’t have sufficient documentation to report this in rigorous manner. We have added this aspect as limitation in the discussion section.

4. c. Did the authors look at early dislodgment (within 5-7 days)? This is another complication related to choice of tube and sometimes position of tube.

Authors’ response: Many thanks for this comment. We don’t have cannula dislodgements during the ICU stay. We have added this complication in the methods. 

5. d. Please list in the methods section all of the specific complications that you looked for.

Authors’ response: As requested, we have updated the list of complications in the methods section. 

6. 2. A critical element of safety is transmission to healthcare providers. This is a very useful element of this study, particularly since the providers had no negative pressure rooms and used only N95s.

a. Please clarify that Powered Air Purifying Respirators (PAPR) were not used.

Authors’ response: As requested we have declared, in the methods section, that PAPR were not used.

7. b. The authors state that no healthcare staff were infected. However, in my experience this is often loosely defined in studies. For example, do you mean just the proceduralists (intensivists, ENT surgeons, anesthesiologists and assistants) or do you include all nurses, technicians and respiratory therapists in the room. If you included all nurses etc, did you survey 100% of the nurses present in the ICU room during tracheostomy to confirm that none developed symptoms? This seems unlikely given this was retrospective study. Please clarify. If you only have information on proceduralists, please state this clearly, and separately mention as a limitation in the limitations paragraph.

Authors’ response: Many thanks for this comment. As indicated in the text, we refer to proceduralists. In our ICU, during the COVID period, we have only 4 doctors and 3 nurses with confirmed COVID-19 disease (mild forms). None of these was involved in the tracheostomy procedures. As requested, we have added this comment in the discussion section.

8. 3. More information on the overall protocol or strategy toward tracheostomy would be useful. Did the unit have a written protocol on timing, procedure, PPE etc?

Authors’ response: Many thanks for this comment. Unfortunately, we don’t have a written protocol. During the Covid period, we have performed bedside tracheostomy as indicated in the methods section. 

9. 4. Why was follow-up truncated at ICU discharge? Since this is a safety-focused study, I would expect information on complications (such as bleeding, infection, granuloma, early dislodgment, need for revision) at least for the duration of hospital admission.

Authors’ response: Many thanks for this comment. Unfortunately, our study, considering the very hard work imposed by COVID pandemic, was limited to the ICU period. We further specify this aspect in the methods section and the discussion section.

10. 5. Information on decannulation would be ideal, however, it is understandable if this is unavailable- please mention under limitations.

Authors’ response: We thank the reviewer for this comment. At the moment of the article submission to PLOSONE, all patients discharged alive from the ICU has been decannulated outside of the ICU (generally in the intermediate care unit or in the rehabilitation department). Unfortunately, we did not collect this data (ex. timing of decannulation). We have added this information in the results section and as limitation.

11. 1. There are now many published studies on tracheostomy performed for COVID-19. Just to mention a few of these- PMIDs 32749607, 32741194, 32709307, 32656673, 32541213 and 32339508. Please describe what this study provides that prior studies do not, and contrast your findings to theirs.

Authors’ response: We thank the reviewer for this comment. As requested, we have mentioned these studies describing the special features of our study.

12. 2. Please discuss why you did not wait 10-14 days, when viral load is lower and risk of transmission to healthcare providers lower?

Authors’ response: As requested, we have discussed this aspect. As indicated in the text, in our daily ICU clinical practice, early tracheostomy was the standard of care even before the COVID-19 pandemic. Therefore, we decided to continue our practice because it was expected to have a long time of mechanical ventilation and, by improving patients’ comfort, reducing the need for sedation and providing a faster weaning from mechanical ventilation. In our hospital, we created “intermediate care” units, which were skilled in the weaning management of tracheostomized patients; this represents an important issue for a rapid access to the post-acute care and also to have more ICU beds available for upcoming patients. Considering these aspects and weighing the risks and benefits, we didn’t wait 10-14 days. Moreover, considering the results, whether the an early timing would increase the risk of transmission because of the high viral load14, it has not been confirmed as none of the participants had COVID-19.

 

ADDITIONAL REQUIREMENTS

Authors’ response: done. 

2. Thank you for including your ethics statement: "Comitato Etico AVEN 562/2020/OSS/AOUPR written informed consent"

Please amend your current ethics statement to confirm that your named institutional review board or ethics committee specifically approved this study.

Authors’ response: done. 

3. Please include the date(s) on which you accessed the databases or records to obtain the data used in your study.

Authors’ response: done (in the methods section).

"NO"

a. Please clarify the sources of funding (financial or material support) for your study. List the grants or organizations that supported your study, including funding received from your institution.

d. If you did not receive any funding for this study, please state: “The authors received no specific funding for this work.”

Authors’ response: The authors received no specific funding for this work. We have included our amended statement in the cover letter.

---

## [Editor Report · Decision Letter 1]

18 Sep 2020

Safety of bedside surgical tracheostomy during COVID-19 pandemic a retrospective observational study

PONE-D-20-24276R1

Dear Dr. Picetti,

We’re pleased to inform you that your manuscript has been judged scientifically suitable for publication and will be formally accepted for publication once it meets all outstanding technical requirements.

Kind regards,

Corstiaan den Uil

Academic Editor

PLOS ONE
---

## [Editor Report · Acceptance letter]

22 Sep 2020

PONE-D-20-24276R1 

Safety of bedside surgical tracheostomy during COVID-19 pandemic: a retrospective observational study 

Dear Dr. Picetti:

I'm pleased to inform you that your manuscript has been deemed suitable for publication in PLOS ONE. Congratulations! Your manuscript is now with our production department. 

Kind regards, 

on behalf of

Dr. Corstiaan den Uil 

Academic Editor

PLOS ONE